# Human Health Benefits of Non-Conventional Companion Animals: A Narrative Review

**DOI:** 10.3390/ani13010028

**Published:** 2022-12-21

**Authors:** Luke Macauley, Anna Chur-Hansen

**Affiliations:** School of Psychology, Faculty of Health and Medical Sciences, North Terrace Campus, The University of Adelaide, Adelaide 5005, Australia

**Keywords:** human–animal bonds, health benefits, companion animals, exotic animals

## Abstract

**Simple Summary:**

Bonds with companion animals are believed to benefit human physical, psychological, and social health. Most research into the human–animal bond has focused on dogs, cats, and horses. However, many people have close relationships with other species such as fish, birds, and reptiles, yet there is limited research into human health benefits that relationships with these animals might offer. This review identified empirical studies and newspaper articles which examined health benefits from bonds with any animal species that was not cat, dog, or horse. We found studies on the health benefits of bonds with companion birds, fish, snakes, tortoises, insects, and amphibians. We also found media articles discussing the health benefits of bonds with rabbits and rats. Studies were primarily descriptive and media articles anecdotal. Nevertheless, the results suggest that non-conventional companion animal species do benefit human health. Further research is needed that examines the human–animal bond, including non-traditional companion animal species, drawing on rigorous empirical methodologies.

**Abstract:**

Research investigating health benefits from household human–animal bonds has focused mostly on bonds with companion dogs, cats, and horses. Wellbeing benefits associated with other companion animal species such as birds, fish, and reptiles are described and anecdotally reported, but there is little empirical literature supporting this. The literature suggests that health benefits of companion animals are predicated on human perceptions of the animal rather than the animal’s species. Therefore, relationships with non-conventional companion animals of diverse species may benefit the health of their human guardians as do dogs, cats, and horses. This narrative review summarizes the current literature exploring perceived health benefits gained from non-conventional companion animals. Searches were conducted for published literature and grey literature up to October 2022 across PsycINFO and PubMed databases, and Newsbank media database for commercial media publications. Nineteen studies and 10 media articles were included in the review. Gaps in the literature include a lack of rigorous research to investigate the health benefits of non-conventional companion animals. Non-conventional companion animals may benefit their guardians by providing social support through acting as attachment figures, facilitating social opportunities and daily routines, fulfilling cognitive needs, and recreating restorative capacities of mindfully observing natural landscapes. Further high-quality research into the human-non-conventional companion animal bond is warranted.

## 1. Introduction

The possible physical and psychological health benefits that people may derive from human–animal bonds has been the focus of numerous studies. However, empirical evidence of health benefits is mixed, with numerous authors pointing over decades to methodological weaknesses in research on the human–animal bond [1,2,3,4,5]. Rodriguez and Herzog [4] attributed one reason for the high variability in human–animal bonds research methodology to the diversity of species and relationships being examined. In contrast to this, one criticism is the field’s focus on dogs, cats, and to a lesser extent horses [6]. These animals are domesticated, trained, and highly accessible for study due to their popularity in many countries [6,7,8,9].

Conclusions drawn from human–animal bonds research are limited when predominantly domesticated mammalian companion animals are sampled, which means meaningful comparisons with or generalizations to bonds with other species cannot be reliably made. Additionally, it is unclear whether the mechanisms by which animals confer benefits to their companion humans are related to the animal species or to other factors such as human perceptions of benefits or otherwise. In the USA, Britain, and Australia, a considerable minority of people care for companion animals of other species, including but not limited to birds, reptiles, rabbits, and fish [10,11,12]. In Canada, rats are increasingly kept as companion animals, but research on companion rats is scarce and focuses on rats’ health rather than human benefits [13]. The popular status of dogs, cats, and horses can be traced to the economic value and practical utility which they have historically contributed to human society, through assisting early humans in hunting, decreasing rodent populations, and providing transport respectively [14,15]. Historical and evolutionary frameworks do not explain the prevalence of other companion animals which do not contribute immediately apparent societal or economic utility, such as birds, rodents, and reptiles. For the purposes of this review, any species which are not domesticated mammals (specifically dogs, cats, and horses) will be defined as “non-conventional companion animals”, focusing this inquiry onto species largely neglected by human–animal bond research.

### 1.1. Theories Applied to Human–Animal Bonds Research

Attachment Theory and, to a lesser extent, Social Support Theory are two dominant frameworks used to analyze and interpret human health benefits associated with human–animal bonds [16,17,18]. Attachment Theory situates animals as attachment figures that confer wellbeing to their human companions by providing psychosocial support through social interactions [19]. Social Support Theory frames companion animals as enhancing wellbeing by providing protective factors against loneliness and isolation [17]. Both theories originated in studies of health in human–human relationships, and may therefore be limited to fully conceptualize and explain the processes of health benefits in human–animal relationships [20]. Furthermore, both theories implicitly depend on humans physically or socially interacting with companion animals, and/or perceiving them as human-like, to explain health benefits derived from the bond.

There are companion animal species less often perceived as human-like and with which many humans tend not to interact physically or socially, such as birds, fish, and reptiles. This raises the question of whether bonds with non-mammalian species also result in health benefits, and if so, by what process? Beck and Katcher [21] proposed Wilson’s Biophilia Hypothesis [22], which states that humans are evolutionarily predisposed to benefit from proximity to nature and animals. Fine [23] has outlined this proposed innate tendency for humans to relate to nature as a mechanism by which the human–animal bond may occur and offer health benefits.

### 1.2. Impact of Animal Species and Human Perception on Human–Animal Bond

There is some evidence that the species of a companion animal is instrumental in the human–animal bond, and consequently in any health benefits. Nielsen and Delude [24] suggested that different species of animal promote different levels of social facilitation, based on their observations of children interacting with dogs, rabbits, birds, and tarantulas. In a study of support provided by companion animals during COVID-related lockdown, companion animal species was significantly associated with the level of support derived from relationships with animals [25]. In that study, bonds were found to be strongest with dogs, cats and horses, but still evident for small mammals, birds, fish, and reptiles. Haddon, et al. [26] found that people developed attachments to companion reptiles within the range of attachment to conventional companion animals such as dogs, but that this attachment varied across species: participants were most attached to companion lizards, followed by snakes and lastly by tortoises. An unpublished dissertation qualitatively explored older adults’ motivations for keeping nontraditional companion animals and found that loving emotional bonds were formed with rabbits, birds, goats, snails, fish, and tortoises [27].

Human perceptions of animals and perceptions of the human–animal bond have been demonstrated to impact on human–animal relationships. Eddy [28] suggested that “morphological and/or behavioral expressions of artificially selected traits are not responsible for the stress buffering effect of companion animals’ presence on human cardiac measures”, and instead nominated the human’s perceptions of the animal as the determining factor in human health benefits derived from the bond. Anthropomorphism is one of these factors, defined as the attribution of human-like intelligence, emotions, and personalities to animals or other non-human agents [29]. Extent of familiarity with and exposure to particular animal species and specific individual animals also have a bearing on human–animal bonds. The level of an individual’s familiarity with a specific animal species is correlated with more compassion for those species and higher belief in that species’ ability to think and feel [30,31].

People live with companion birds, reptiles, rabbits, and fish in numerous countries, including the USA, Britain, Australia [10,11,12] and Canada. Health benefits associated with these non-conventional companion animals and others are widely anecdotally reported—for example, on 8 February 2022, a Google search for “health benefits of pet fish” returned 34.0 million results, while a search for “health benefits of pet birds” returned 16.5 million results. Yet there is a distinct lack of empirical research in these areas. Furthermore, existing research on human interactions with these animals tends to explore risks and harms, such as exposure to Avian Flu through poultry [32] and exposure to salmonellosis through companion reptiles [33].

Research into bonds with diverse companion animal species would provide a more complete view of the role different species of animals play in human health and lives, help determine the specific mechanisms by which different companion animals facilitate health benefits, or not, and enable meaningful comparisons to be made across different species. This review examined the current state of research into the reported wellbeing benefits provided by non-conventional companion animals living in their human guardians’ households. In doing so, this review aimed to summarize what is known regarding wellbeing benefits of non-conventional animals and identify areas where further research is needed to ascertain possible health benefits of companion animals, more broadly defined than cats, dogs and horses.

## 2. Materials and Methods

A narrative synthesis methodology was chosen because this method is appropriate in areas of sparse research, and is suited to integrating results of studies with highly variable methodologies [34]. The first author classified findings from empirical studies by categorizing them into overarching themes [35,36].

All peer-reviewed publications pertaining to human health benefits derived from bonds with household companion animals other than dogs, cats, or horses were included. Grey literature and commercial publications were also included to widen the available material to include for review. The timeframe for included articles was not limited. The end date for included searches was October 2022.

Eligibility criteria were that studies must be (1) written in the English language, (2) relate to human health benefits, and (3) frame these benefits as effects of or related to human bonds with domestic household animals. Exclusion criteria were (1) studies which examined health benefits relating to bonds with companion dogs, cats, or horses, and (2) studies analyzing the effects of Animal-Assisted Interventions or Therapies. The latter criterion focused the review on the benefits of personal, everyday relationships with household companion animals. These interactions differ considerably from the structured, clinical contexts in which Animal-Assisted Therapies take place, and in which the human–animal bond plays a crucial but complementary role to a trained therapist [37].

The literature search was conducted in June 2021 and again in October 2022, and encompassed PsycINFO and PubMed databases, and Newsbank media database for commercial media publications. Supplementary searches were conducted over Google and Google Scholar to verify the reach of prior searches, and ensure all relevant studies were captured. Owing to the broad scope and the highly variable reporting of companion animal species in human–animal bonds research, search terms were applied to the title field and abstract field, with the words “dog” and “cat” excluded. A methodology quality assessment procedure was not carried out for the present study due to the wide range of study designs included. No date range was stipulated (Appendix A).

Search terms (see Appendix B) were developed in collaboration with a research librarian and used for the PsycINFO search. Search terms comprising different species of animal were developed through preliminary searches of scientific and commercial literature to identify animals that may be kept as companions. Specific parameters were refined over multiple searches to filter out studies of dogs, cats, and horses while retaining broader human–animal bonds studies. PubMed and Newsbank searches followed an equivalent search protocol using the same terms. Searches across all three databases used (1) one identifier for health or quality of life benefits to humans, (2) one identifier for the human–animal bond or interaction, and (3) one identifier for different species of animal, excluding subject headings relating to dogs or cats. A complete list of search terms used is available in the Appendix B. Searches were repeated with the species identifier omitted, but with the identifier excluding dogs and cats retained. Reference lists from included studies were consulted to determine whether any appropriate publications had been missed. See Appendix B for the PRISMA flow diagram.

## 3. Results

The literature search yielded 19 articles, 10 commercial media articles, and 1 unpublished thesis, which examined motivations for and benefits of keeping companion animals in the household that were not dogs, cats, or horses. The 19 included studies are summarized in Table 1. Animal species represented in this review are birds including parrots and chickens, reptiles including tortoises and snakes, as well as amphibians, fish, and crickets. Seven overarching themes describe the possible processes or mechanisms by which companion animals may benefit their human guardians’ health: (1) Companionship and Attachment, (2) Social Facilitators, (3) Purpose and Routine, (4) Connectedness with nature, (5) Decoration and Aesthetics, and (6) Physiological Benefits. There was an additional theme for (7) Commercial Media, in which human health benefits of companion animals were discussed in non-scientific contexts, and explanatory mechanisms of health benefits were not evident.

Of the 19 studies reviewed, two were randomized controlled trials and one was a non-randomized controlled trial. There were five surveys, two of which were conducted during scale development. There were four mixed methods studies employing a combination of observation, interviews, and surveys. One of these also included an ethnographic component, while a separate study was entirely ethnographic. Of the remaining studies, there was one case study, one qualitative interview study and three reviews, two of which were narrative reviews with another being an Editorial.

Recruitment in the selected studies was often conducted through convenience sampling and snowball sampling within niche hobbyist groups, the results of which may be distorted by selection biases. This reduces the representativeness and generalizability of findings as the people most likely to participate were likely those who find unique companion animals most rewarding to keep. Furthermore, quantitative studies were limited in representativeness by generally low numbers of participants, and by recruiting participants from within specific locales. Representativeness of samples and the ability of authors to generalize findings more broadly are further limited by human–animal bonds being highly culturally bound [56], and by animal protection and conservation laws dictating which species of companion animal can be legally kept which differ by country. Most studies were conducted in the United States at a total of 11, with one conducted in Australia, one in Germany, and two in Korea. The three review studies appear to have drawn only from studies written in English.

Both qualitative and quantitative approaches tended to be descriptive and used participant self-report and/or observation to collect data, which can be vulnerable to confirmation bias [57]. Research of health benefits from bonds with non-conventional companion animals is dominated by such approaches, and therefore cannot ascertain causation, direction, or strength of health benefits, or generalizability to other populations. This review identified several notable exceptions. One is the well powered study conducted by Burmeister used to develop a scale of people’s relationships with their companion birds which recruited n = 1444 participants across Germany [45]. The pair of studies conducted by Ko, Park, and others, included a single-blinded Randomized Control Trial, and a functional Magnetic Resonance Imaging study, on the effect of caring for companion crickets on brain function and overall health [49,53]. In an earlier study, Riddick [54] conducted a non-randomized controlled trial in which participants were given aquaria stocked with fish to keep at home, and which measured outcomes across a variety of social, psychological, and physiological health indicators. Future research of bonds with non-conventional companion animals may use the example set by these studies as they demonstrate viable methods of empirically testing the benefits of companion animals.

### 3.1. Companionship and Attachment

The most consistently reported benefit of keeping any species of companion animal was the companionship provided by the animal, with a total of 17 included peer-reviewed studies reporting that companionship provided by and attachment to companion animals were benefits of bonds with companion animals. Anthropomorphism of companion animals was frequently reported alongside accounts of animals providing companionship. This suggests a possible correlation between anthropomorphism of non-conventional animal companions, and the levels of social support and other health benefits gained from the human–animal bond. However, no studies reviewed tested this connection.

#### 3.1.1. Companionship and Attachment with Birds

Companion birds’ high intelligence and ability to recreate human speech have been reported to enhance the sense of companionship people feel with them [42]. In Australia and the USA, birds are the third most popular companion animal behind dogs and cats, and in the United Kingdom are fourth most popular behind dogs, cats, and rabbits [10,11,12]. Corresponding with their moderately popular status as companion animals, the body of literature examining human–bird relationships is relatively large compared to other non-conventional companion animals. However, even this body of research is somewhat niche in the scheme of human–animal bonds literature—in a systematic review of human–animal research, birds were only discussed in reference to poultry being kept as agricultural livestock rather than companion animals [6]. Additionally, scientific inquiry into human–bird relationships has tended to use or adapt scales developed for use with dogs and cats, which are likely to miss the nuances inherent in a human–bird relationship owing to fundamental differences in species, daily care and veterinary care, and birds’ cognitive and speaking capabilities [45].

In 1988, Altman published one of the earliest peer-reviewed articles describing human–bird relationships outside of Animal-Assisted Intervention contexts [39]. She argued that human–animal bonds research neglected companion birds in favor of studying human bonds with cats and dogs. As well as summarizing studies of Animal-Assisted Interventions using birds, Altman’s review looked at the benefits of informal human relationships with birds outside of structured interventions by drawing from media and anecdotal accounts. She also referenced personal communication with Beck and Katcher, the authors of a survey which investigated the roles companion animals play in family structure [39]. Altman reported that in her correspondence with Beck and Katcher, they divulged that participants with birds were not statistically different to those with cats on levels of attachment when the results of participants with birds were analyzed independently of those with animals included in the “other” category. Although Altman’s review did not include direct testing, she established a case for future studies to research wellbeing benefits associated with attachment to companion birds.

The following year, Beck and Katcher [42] conducted surveys and interviews in the United States comparing people with birds (n = 42) against results previously obtained from participants with companion cats and/or dogs on results of the Pet Attachment Survey [58]. The study also included observations of interviews with participants as they interacted with their companion birds (n = 18). The results, being the first record of human–bird relationships specifically studied in comparison with relationships with other companion animal species, described the potential for reciprocal affection and deep attachment in human–avian relationships. Participants treated companion birds like family members at a rate of 90%, more frequently than cats at 73%. Furthermore, participants with birds were as likely to talk to their birds as were those with dogs and/or cats. The authors also noted that participants found talking with their birds to be highly rewarding, were capable of sustained dialogue and interaction with their birds, affectionately touched their birds, and held their birds at eye contact to facilitate face-to-face connections. Based on their findings, Beck and Katcher [42] argued that participants received wellbeing benefits from companionship provided by birds “through which the pet exerts a positive influence on health”. This “positive influence”, they proposed, was because someone reduces “his own state of activation to sustain the dialogue” with a companion bird [42]. Here, birds are distinct from cats or dogs which are larger and not hollow-boned, and therefore less vulnerable to injury from mishandling. Although an interesting and novel hypothesis, the exact mechanism of these health benefits was not examinable by the study methodology and was left as an avenue for future research.

Kidd and Kidd [47] conducted a study in the United States of n = 100 participants in which they measured reported benefits and disadvantages of living with companion birds, and personality characteristics of people with companion birds. Of those aspects of birds that participants enjoyed most, companionship provided by birds was rated most highly at 38%. Companion birds’ dialogue with the participants and with other birds was also highly rated at 28%, while lower-rated but still relevant reasons given included birds’ friendliness (10%), receiving love from birds (7%), and feeling relaxed with birds (4%). A sizeable portion of respondents (36%) had adult children no longer living in the home while 25% were single and childless, which in conjunction with participants reporting enjoying companion birds’ company further supports Anderson’s statement that companion birds alleviate loneliness and the “empty nest syndrome” in family homes [40]. Kidd and Kidd concluded that human–avian bonds “can often be more warm and caring than human interactions with a dog, cat, or horse” [47]. They attributed this to birds’ high level of intelligence and resulting ability to meaningfully interact with humans such as by learning tricks, as well as some bird species’ ability to speak meaningfully with humans. This all relates to aspects of avian behavior and cognition which participants may have perceived as making birds more human-like than other companion animal species. As previously discussed, animals which are considered more human-like receive more compassion from humans, and this compassion may facilitate stronger bonds and health benefits [59]. However, given Kidd and Kidd’s methodology including the use of self-report data collection, their conclusions must be considered as conjectural—they did not compare attachment to or perceptions of different companion animal species as more or less human-like within or across individual participants. Likewise, they did not compare participant perceptions of or attachment to bird species capable of speech with those that were not.

Loughlin and Dowrick [51] surveyed 80 people in Alaska about their motivations for having birds. Reasons were aggregated into motivators fulfilling social, esteem, cognitive, safety, power, and aesthetic needs. Results showed that companion birds most frequently fulfilled social needs, followed by esteem and cognitive needs. Respondents rated prominent factors of their relationships with their birds as including expressing affection for their birds (83%), their birds’ personality (75%), and friendship and companionship provided by birds (70%), while fun and laughter provided by the bird were important for 59% of respondents. For 60% of respondents, social needs were satisfied by talking to, feeding, grooming, or watching their birds, and 59% reported that their birds added more meaning to their lives. A further 52% of participants rated feeling needed by their birds as an important motivator for living with birds. These findings indicated that birds may enrich their keepers’ lives, primarily by fulfilling social needs through providing affection and enjoyment. This study identified the importance of a bird’s personality to the human–bird relationship and emphasized birds’ vocal abilities as also distinguishing humans’ relationships with companion birds from other companion species.

#### 3.1.2. Fish as Companions

Riddick [54] conducted a non-randomized control trial in which she tested the effects of giving aquaria stocked with companion fish to elderly people living in low-income subsidized housing on blood pressure, leisure satisfaction, and relaxation states. This study has been included in the review as the fish were introduced in a domestic setting without the structure of formal treatment, allowing potential human-fish bonds to form and distinguishing this from Animal-Assisted Interventions. The total sample size was 22, with seven participants in the experimental condition given aquaria, eight participants who received scheduled visits as a comparison group, and seven participants in the control group. Interestingly, while the aquarium group underwent a significant increase in relaxation, the visitor group showed a significant decrease in loneliness compared to the other two groups. These findings suggest that fish as companions may confer wellbeing benefits to their human guardians, but that companionship provided by fish is not as beneficial as that provided by humans.

Kidd and Kidd [48] also conducted a study using a similar methodology involving surveys and interviews of n = 100 people with fish in home aquaria. The survey measured participants’ attitudes and feelings about their fish, and in what household roles they placed their fish. Participants were also asked to list benefits and problems associated with keeping fish. The authors found that 72% of participants considered their fish to be “family pets”, while 4% thought of their fish as “companions”. The authors did not explain their decision to distinguish offering survey items as fish as pets versus fish as companions, but this option appears not to have been offered for the same authors’ similar survey conducted with participants with household birds [47]. The authors also did not elaborate on the finding that far more participants considered fish to be “pets” rather than “companions”, but the difference seems to have been salient to the respondents. It is possible that fish were perceived as being the objects which were the focus of a hobby rather than anthropomorphized companions, which is supported by research in which participants rated goldfish as “Not at all similar” to humans [29].

Langfield and James [50] interviewed nine people with home aquaria based in Australia in which they explored participants’ motivations for and the benefits of keeping fish. They adopted a phenomenological study design in response to the absence of literature exploring motivations for keeping fish as companion animals. Participants were recruited using snowball sampling within the authors’ social circles related to the hobby of keeping companion fish. Participants discussed attachment to their fish in the context of different responses to the deaths of companion fish. Some participants reported flushing dead fish down the toilet, which the authors posit indicates lower attachment. Another participant recounted, “I don’t name my fish anymore because you get attached and then they pass away” [50]. Another participant regretted that her fish “don’t care” about her and that she could not “talk to [her fish] as you would to a dog or a cat” [50]. However, other participants indicated higher levels of attachment to their fish, with one sharing that she abstained from eating seafood because it reminded her of her own companion fish. Some participants reported enjoying the company of and talking to their fish. These findings suggest the capacity for fish to provide companionship and act as attachment figures is highly variable and may be associated with perceptions of fish as thinking, feeling creatures rather than decorations. The authors suggested that attachment to companion fish, or indeed any companion animal, is related to the level of interaction possible with the species—fish, being confined to aquaria, are minimally capable of interaction.

#### 3.1.3. Companionship and Attachment with Herptiles

The status and practice of keeping herptiles, the family of animals including reptiles and amphibians, as companion animals is contentious owing to perceptions of herptiles as fearsome, disgusting, and dangerous [60], and as lacking intelligence and emotion [61]. Yet multiple studies in this review described close personal and beneficial relationships with herptiles. Eddy’s [28] case study of a participant’s blood pressure decreasing while handling his companion snake, represents an example of physical health benefits facilitated by a non-conventional companion animal. Eddy suggested that attachment to the snake was an important driver of stress alleviation, but not the only factor, and states that the “data show clear parallels to the cardiac responses of dog owners to their dogs” [28]. As a case study, the findings of Eddy’s findings are not generalizable, and do not provide detail as to the frequency of this effect in human–snake bonds. However, the study shows that physical health benefits gained from bonds with snakes are possible. In the absence of wider human–snake bond research, the extent of this effect cannot be definitively stated, further reinforcing the need for human–animal research encompassing more diverse species.

Kampfer and Love [46] published a study detailing the development and results of a Tortoise Caregiver Questionnaire, based on surveys of 490 Vegas residents with companion desert tortoises. The study aimed to assess participant motives for obtaining and keeping their tortoises. The questionnaire items mapped onto the following domains adapted from the Pet Attitude Scale: aesthetic; anthropomorphism of the tortoise; sharing tortoise care duties; entertainment provided by the tortoise; feeling needed by the tortoise; escape from daily stresses; companionship; duty to the tortoise in fulfilling its needs; and relaxation associated with interacting with or watching the tortoise [62]. Kampfer and Love also added items measuring interaction with tortoises, adapted from a study on feeding wild birds in household backyards [63]. They opted to exclude items from Templer’s Pet Attitude Scale which measured interactions through love but retained items measuring physical interactions. They do not provide any reason for doing this, but the decision may have been informed by anecdotal accounts of people caring less about tortoises. The Companionship domain ranked seventh out of ten domains, while Interaction ranked last. However, statistical analyses showed that Companionship and Interaction, while not as impactful as other domains, were still of considerable importance to participants. Therefore, benefits relating to companionship were important but not primary factors which motivated participants to keep their tortoises [64].

Azevedo [38] published a mixed-methods study examining 220 reptile keepers’ motivations for acquiring companion reptiles and perceptions of their reptiles’ sentience. Companion reptiles were most frequently defined as family members (by 64% of participants), followed by 43% viewing their reptiles as pets and 21% as friends. These views and apparent level of attachment differed by species, with lizards reported as being family members most often, followed by chelonians (turtles, tortoises, and terrapins), and lastly snakes. Over half (54%) of participants reported terms of affection such as “like” and “love” as the primary motivators in their decision to acquire companion reptiles. Among qualitative responses, 22 participants volunteered “companionship” as a benefit of keeping reptiles. One participant described his tortoise as conveying calmness and affection. The authors concluded that attachment with companion reptiles is possible, but that human-reptile relationships are more nuanced than bonds with conventional companion animals, for example, by including aspects of intellectual curiosity and fascination.

In general, companionship and attachment seem associated with the level and type of interaction in which someone engages with their companion animal. This is corroborated by Muldoon et al. [52], who examined correlations between attachment and psychosocial benefits in a large dataset of 6700 children across a diverse range of species including dogs, cats, fish, reptiles, and amphibians. Measures of interaction and reciprocal communication were fundamental to their analysis and were highest with companion dogs. They found that attachment to companion dogs was highest and was associated with psychosocial benefits for children. Attachment to other species was not associated with psychosocial benefits, except for attachment to companion cats, which was also associated with negative effects on quality of life.

Birds can engage in reciprocal physical affection and mimic human speech, while fish and tortoises are sequestered in enclosed habitats. People may still talk to and show physical affection for other companion animals which are less capable of reciprocal interaction, such as tortoises, snakes, or insects. Azevedo [38] found that participants’ perceptions of their companion reptiles’ attempts to communicate were significantly correlated with species, with 54% of participants reporting lizards attempted to communicate and 49% of those with chelonians, whereas only 20% of those with companion snakes reported any attempt to communicate. In the absence of interaction (reciprocal or otherwise), motivators other than companionship and attachment must be considered to determine the drivers of keeping non-conventional companion animals, and the possible wellbeing benefits.

### 3.2. Social Facilitators

As well as or instead of being attachment figures which provide companionship, household animals may facilitate interactions with other humans, effectively acting as catalysts to provide benefits of social interaction to their human guardians. Both traditional and non-traditional companion animals can prompt conversations with other humans. However, non-conventional companion animals may prompt people to join niche hobbyist groups, to discuss particular care needs for their companion animals and the intricacies of maintaining dedicated habitats or enclosures.

In Kidd and Kidd’s [47] survey of motivations for keeping birds, of those participants who lived with their children, approximately a third reported that their children helped care for their companion birds, though the remainder tolerated, feared or were indifferent to them. In these circumstances, companion birds may act as social facilitators within families, improving relationships between family members. Companion birds have been noted to benefit wellbeing by acting as social facilitators through, for example, providing shared hobbies and topics of discussion between individuals who may otherwise suffer loneliness or alienation, similar to other companion animal species such as dogs, cats and rabbits [65,66,67].

### 3.3. Purpose and Routine

Engaging in routines necessary to care for non-traditional companion animals, which often require specialized food, habitats, and veterinary care, can provide people with a sense of meaning and purpose. Langfield and James [50] identified the importance of routine and purpose for participants with companion fish, whose tanks required consistent and careful maintenance on a regular schedule. Similarly, Loughlin and Dowrick [51] found that 59% of participants reported that their birds added more meaning to their lives, while 52% rated feeling needed by their birds as an important motivator for keeping them. Ko et al. [49] found 78.3% of participants self-reported that their companion crickets “were cared for with careful attention and regular feeding”. The authors did not discuss the possible role of routine or feeling needed in benefitting participants’ mental health, but there appears to be a link. Similarly, in an fMRI study of cricket-rearing’s effect on brain function, executive performance was improved in the cricket-rearing group which may have related to participants’ routine care for companion crickets [53]. Azevedo [38] found that ‘duty of care’ was an important motivator for participants keeping companion reptiles, which may also relate to themes of Purpose and Routine represented in other studies.

### 3.4. Connectedness with Nature

Non-conventional companion animals often require dedicated enclosures such as aquaria or chicken coops to cater for their more specific living and care needs. To recreate the natural habitats of such animals, these enclosures might include specific lighting, water features, and arrangements of plants, rocks, and specific soils. By virtue of both these enclosures and perceptions of some non-conventional companion animals as more wild and natural, these animals may facilitate a sense of connectedness with nature for their guardians. Blecha and Leitner [43] published a study of urban residents with backyard chickens in Seattle and Portland, USA, and found that passively observing chickens had mentally restorative benefits and alleviated stress for participants. One participant recounted how her chickens’ clucking was “soothing” and that watching them was very “stress-relieving”, while another reported that being around her chickens was a “zen, peaceful” experience [43]. Experiences which portray companion chickens as being stress-relieving differ markedly from other accounts explored in this review wherein humans seem to receive wellbeing benefits from actively interacting with their companion animals as attachment figures which alleviate loneliness. Blecha and Leitner related this finding to prior research identifying “soft fascination” and “being away”, which describe aspects of natural landscapes that are mentally restorative for people after periods of stress or hard concentration [68,69]. Based on this, it was proposed that “Chicken spaces are islands of calm within the city” which are “psychologically distant from [their participants’ often stressful] nonagricultural workplaces” [43]. While the study methodology did not allow direct testing of this idea, it has face validity as a possible explanation for explaining the wellbeing benefits of non-traditional companion animals with whom humans tend not to interact but instead tend to passively observe.

Elements of non-conventional companion animals conferring wellbeing benefits through “soft fascination” and “being away” relate to and may explain the findings of other studies. In noting that their participants found watching companion birds to be relaxing, Loughlin and Dowrick [51] suggested that watching companion birds may have health benefits in its parallels to similar calming activities such as birdwatching in the wild and watching fish in aquaria. Later studies found that birdwatching reduced participants’ tension and fatigue [70], and that neighborhood bird abundance is associated with lower population depression, anxiety, and stress [71]. “Soft fascination” may also explain participant accounts from Kidd and Kidd’s study [48] claiming that home aquaria helped reduce their blood pressure.

Kampfer and Love’s [46] survey of people with companion desert tortoises also provides some evidence that people find passively observing non-conventional companion animals to be rewarding and a significant motivating factor in keeping them. Tortoise factors relating to Aesthetics were the second-most reported reason for keeping tortoises, and the Escape domain is also possibly related to the feeling of “being away” prompted by passively spending time with companion animals. For example, one participant stated that “a half hour of tortoise watching is better than medicine” [46]. The authors suggested that desert tortoises “appeal to those who wish to relate more strongly with the natural elements of the desert and are a reminder of that aspect even in an urban environment” [46], and provide as evidence a prior study which explored how humans can increase “connectedness with nature” through companion animals [72]. Blecha and Leitner’s [43] concepts of “soft fascination” and “being away” may relate to Kampfer and Love’s concept of “connectedness with nature” provided by non-conventional companion animals, particularly those which require dedicated habitats or enclosures owing to their unique care needs. These spaces, constructed specifically to replicate the natural habitats of the non-conventional and sometimes exotic companion animals they contain, double as small representations of natural landscapes for the animals’ human guardians. More consistent and consolidated terminology is required in human–animal research to allow for higher levels of evidence gained through meta-analyses, for example, as evidenced by conceptual overlap between terms such as connectedness with nature and soft fascination.

According to Kidd and Kidd’s [48] study of motivations for and benefits of keeping home aquaria, companion fish are another species from which human guardians may receive wellbeing benefits through passive observation. Of 100 participants, 70 reported that their fish “calmed them, helped them to relax, and reduced their stress” [48]. The authors further suggested that “Perhaps the major motivation is the reported calming, relaxing, stress-reducing effects of watching fish” [48].

In discussing the results of their randomized control trial testing companion crickets as an intervention for improving mental and physical wellbeing, Ko et al. [49] stated that they selected the specific species of cricket used because it was common to the East Asian region, and that “the association between cricket chirping and rural life may cause the elderly subjects to feel nostalgic and therefore affectionate towards the insects” [49]. This may also be framed as connectedness with nature in leveraging crickets’ chirping as associated with nature and rural life to cause or supplement wellbeing gains. However, they did not undertake specific testing of this idea, nor did they report any participant accounts of nostalgia linked to crickets’ chirping.

### 3.5. Decoration and Aesthetic

It is possible that people benefit from the appearance of non-conventional companion animals and their specialized habitats simply because both act as decorative points of interest which decorate the household. Kampfer and Love [46] measured their participants’ Aesthetic motivations to keep companion tortoises through the items “I feel that tortoises add beauty to the environment” and “I think the world would be a less attractive place without tortoises”. The items lacked sufficient specificity to determine whether they measured participant notions of tortoises as decorative objects or as facilitators of connectedness with nature.

Companion animals contributing to their guardians’ wellbeing by beautifying and decorating their shared living spaces is exemplified by companion fish kept in home aquaria. Kidd and Kidd suggested that people have home aquaria because they “enjoy the beauty and gracefulness of their fish” [48]. Correspondingly, 22% of their participants considered their companion fish to be “room decorations” [48]. When asked about the benefits of home aquaria, a further 4% answered “Entertainment” as the primary benefit. Similarly, Langfield and James’ participants were “attracted to the different varieties” of fish species available, and one preferred Australian native fish because he considered them to be “more attractive than imported fish” [50], perhaps suggesting moral motivations relating to bioconservation. Another participant reported that native fish species’ “different body shapes, different color patterns and things like that are far more interesting to watch” [50].

### 3.6. Physiological Benefits

Some included studies commented on physiological benefits derived from bonds with studied companion animals. Riddick’s [54] controlled trial testing the effects of goldfish aquaria on participants’ blood pressure and mental states found limited evidence for the physiological benefits of non-conventional companion animals. Participants in the experimental group showed a significant decrease in diastolic blood pressure, however, post hoc tests revealed their diastolic blood pressure was also significantly higher than the control or comparison groups before receiving the intervention. This makes the validity of Riddick’s findings less clear, but they nevertheless contribute evidence to the argument that companion fish can benefit their human guardians’ blood pressure and overall health.

Ko et al. [49] tested the effects of an eight-week cricket-keeping intervention on 94 participants’ mental and physical health. The insect-caring and control groups were healthy and matched on baseline characteristics, and post-intervention tests showed the insect caring group had significantly lowered Geriatric Depression Scale scores and improved Mini-Mental State Examination Scores. No significant differences between groups were found on physical indicators, which the authors attributed to the relatively short intervention timeframe. Their finding that Mini-Mental State Examination Scores improved among people with companion crickets is especially interesting as it shows a benefit to cognitive functioning of companion animals beyond alleviating mental illness, loneliness, or stress.

Ko’s cricket-keeping intervention was followed by a 2019 fMRI study, also conducted in Korea, which recruited 16 older female adults [53]. This study replicated the cricket-keeping intervention detailed previously, with the addition of the experimental group being divided into high- and low-scores on the Wisconsin Card Sorting Test. This test was chosen as a validated test which draws on the use of a distributed brain network, making it appropriate for measuring participants’ overall brain function. fMRI analysis showed increased activation in the right dorsal lateral prefrontal cortex and parietal cortex of insect-rearing participants when the semi-Wisconsin Card Sorting Test was performed, but not during the high-level baseline version. Overall, their testing found that raising companion crickets was associated with increases in executive function and performance improvement in older women. The process which produced these effects was not explicated in the study report, but may be related to the routine care required by companion crickets, and which was regularly checked by an experimenter during weekly compliance phone calls. The findings were not generalizable as the sample consisted of female participants of a specific age drawn from a single Korean city. However, the results do provide possible evidence that participant self-reported benefits of keeping non-conventional companion animals may be supported by measurable cognitive and neurological changes. The application of fMRI study methodology to assess neurocognitive benefits of insect-rearing is highly novel and provides a basis for similar studies with other companion animal species. There is an important cultural element to this pair of studies, as keeping crickets as companion animals is historically more common in East Asian cultures than in the West [73]. This cultural element may further limit the generalizability of Ko and Park’s findings, especially to non-East Asian cultures, but they also demonstrate the importance of culture and individual perceptions of companion animal species on benefits gained from the bond.

### 3.7. Commercial Media Describing Benefits of Non-Conventional Companion Animals

Only ten relevant articles were located through searching Newsbank media database. The included media articles tended to be sparse on detail and published in the tone of human-interest stories, which likely reflect the interests and lay knowledge of broad audiences relating to the health benefits of keeping companion animals.

A 2010 article published in Right Vision News, Pakistan, includes a brief mention of human health benefits provided by backyard chickens from an interviewee, who believed that “her chickens offer her and her husband significant health benefits” [74]. She claimed that she “learned a lot about the nutritional benefits of eating eggs raised in a free-range environment” [74]. No other mention is given to human health benefits, with most of the article focusing on healthy living conditions and optimal care for backyard chickens. Another article, published in the Metro in the UK, claimed that keeping backyard chickens can bring relaxation to households, reduce isolation, and lift spirits, and that caring for them offers a “form of mindfulness” [75]. Chickens are discussed by interviewees with personal affection and by reference to their names. Another stated that “hen-keeping promotes wellbeing” and “reminds [her] of the marvels of life” [75]. The mental health benefits of companion chickens were also discussed in an online article by Rouse published on MSN [76]. Rouse said that backyard chickens “make for amazing companions” and motivate their human companions to “get some Vitamin D and those endorphins flowing” [76]. This indicates that mental and physical benefits of backyard chickens are derived from the physical act of caring for and being around them in an outdoors environment, not only as a direct effect of the human–chicken bond. Rouse further stated that observing chickens is both entertaining and “can have a major calming effect” [76], due partly to chickens having different personalities.

Companion rabbits were the subject of a brief news article authored by Watthanachan and published in The Nation, a Thai newspaper [77]. Watthanachan described the bond between 30 rabbits and their human companion, Nattawut, who stated that “the benefits greatly outweigh the work” [77]. Companion rabbits were reported to help “people rid themselves of stress and worry after long days at work” [77], without providing evidence or elaborating on how these benefits are achieved. The Dayton Examiner, a newspaper based in Ohio, USA, published an article describing the mutually beneficial effects of massaging companion rabbits and other animals [78]. The article states that massaging companion animals reduces the anxiety of both the human and the animal, lowers the blood pressure of both, promotes bonding between the two, and accelerates recovery from surgery or illness [78]. These claims are made without reference to evidence except a promotion of an instructional book on massaging companion rabbits.

An article published in the Sun Journal in Maine, USA, advertises guinea pigs for adoption from an animal shelter [79]. The author espouses the benefits of companion guinea pigs, stating “you’ll feel your stress level go down” and that “guinea pigs are very social animals”. The author also states that guinea pigs “each have their own little personalities”, are “friendly with people”, and “are affectionate and loving pets”. This article portrays guinea pigs as relaxing social companion animals with distinct personalities.

Burke authored an article titled ‘I’m glad the fish died’ which was published in the Alaska Dispatch News [80]. Burke describes “pet guilt” associated with caring for a companion fish purchased then abandoned by her children [80]. She relates that she “never expected to one day grow marginally fond of our pet fish”, that chores relating to the fish’s care became the source of arguments in the family, and that watching the fish’s health decline caused her stress [80]. However, she goes on to describe that without the fish “The kitchen seems lonelier” [80], and that the fish’s death reminded her of “the importance of doing less in daily life” to make more time for mindful engagement with hobbies and loved ones [80]. Burke’s account mirrors the complicated and often ambivalent relationships people have with companion fish described by published literature, in which companion fish deaths might elicit less grief than deaths of companion mammals. As in Burke’s account, companion fish may also confer health benefits to their human guardians by providing peaceful spaces for reflection.

Sherman [81] reported on the benefits of companion betta fish for college students’ stress and anxiety in an article written for the Buena Vista University newspaper The Tack. In this article, Sherman described relaxing by watching his companion fish swim, complaining to his fish about classes and other problems, and teaching the fish tricks. He indicated that betta fish “develop their own personalities”, and that the fish’s “colorful body and tank brighten up [the] little dorm room, that can sometimes get dark and depressing”. Sherman also reported on a conversation with a doctor, who stated that “watching fish swim back is stress and anxiety reducing”, and that “the gurgling sound of the bubbles add to the therapeutic effect of looking at the tank”. This suggests that companion fish confer benefits by decorating their human companions’ houses, as described elsewhere in this review. Sherman also suggested that companion fish are “a great compromise” for students living away from home who miss their other companion animals, situating companion fish as temporary placeholders in the absence of preferred conventional companion animals.

An account of the mental health benefits of companion rats is provided in an article published in the Washington Post titled ‘They’re intelligent and friendly. Why some people think rats are the perfect pet, for fun and comfort’ [82]. One interviewee adopted two rats to help her overcome anxiety and depression. She recounted how the rats “are what made [her] get out of bed, knowing [she] had those little lives to care for” [82], evoking the theme of Purpose and Routine as a mechanism of health benefits. She explained that rats aided her recovery from mental illness because “You might not have the energy to walk a dog, but you can manage to fill a water bottle” [82], making companion rats uniquely suited for providing health benefits over dogs as the more conventional companion animal. An interview with another person with 36 companion rats is included, described as a domestic abuse survivor. She is reported as saying that:

The rats have helped me and my girls cope with a lot of emotional and psychological healing. The boys snuggle under our chins when we have flashbacks, anxiety issues or when we are feeling particularly sad. They help a lot with rebuilding our confidence and quieting our nerves… Recovery from trauma is a long road… and I would say that it is certainly easier with these little guys helping us [82].

Another account of the mental health benefits of companion rats is provided by spouses Cindy Stuart and Phillip Stuart in the July 2010 issue of the Animal Human Interaction: Research & Practice Newsletter [83]. In this article, Cindy drew on her profession as a psychologist to describe the role of their companion rats on her husband Philip’s recovery from pancreatic cancer. Philip also contributed, stating:

As a cancer patient, I feel a lack of control over my life. My own self-worth is battered by the sometimes cold, occasionally demeaning nature of my experiences within the overworked, oncological branch of the medical profession. But it’s not that way when I’m with my rats. They love and respect me unconditionally. They don’t know I’m sick. [83]

Cindy suggested that Philip’s “positive state of mind, in which [their] rats had a huge influence, has not only helped to bolster his biological responses, but has enabled him to thus far lead a more full and normal life”. [83]

News stories describing the mental health benefits of non-conventional companion animals indicate the importance these species have to many people, and the potential benefits they may provide to their human companions’ health. Yet no published empirical studies into the benefits of companion species such as rats and rabbits were identified for this review.

## 4. Discussion

There are numerous animals providing attachment, social support, and links to nature, but for which little, weak or no empirical evidence of human health benefits deriving from these functions exist. For example, the selected articles from commercial media showed anecdotal evidence for rats and rabbits providing mental health benefits to their human companions, but no studies verifying such effects have been conducted. The webpage of Emotional Pet Support states that “All domesticated animals may qualify as an Emotional Support Animal (cats, dogs, mice, rabbits, birds, hedgehogs, rats, minipigs, ferrets, etc.)” [84], but research has not investigated these non-traditional species. There is a divide between lay beliefs in wellbeing benefits of different companion animal species and the empirical literature supporting such beliefs.

Based on this review, there is evidence that non-conventional companion animals are perceived to improve their guardians’ health through a variety of different mechanisms. However, most methodologies relied on self-report measures and did not allow for actual health benefits to be determined. The finding that animals are perceived to provide health benefits by facilitating connectedness with nature, adherence to routines, and/or social interactions with humans show that companion animals may go beyond acting as attachment figures or providing social support. Although animals being capable of physical interaction and reciprocating interaction was an important feature of human–animal bonds in many studies, there was also some evidence that many non-conventional companion animals with whom physical interaction and affection is limited or impossible also enhance their guardians’ health. For these species, passively observing companion animals may produce health benefits through processes similar to mindfulness and meditation. This suggests that reciprocal interaction may not be necessary for a healthy bond to form, or that positive health benefits may also occur through multiple processes, including social mechanisms such as attachment, and psycho-biological processes such as connectedness with nature. While Muldoon et al. [52] only found psychosocial benefits associated with attachment to companion dogs, their survey only tested attachment and focused on communication and interaction as mechanisms. This further suggests other mechanisms may be responsible for health benefits associated with nonconventional companion animals. For example, one specific pathway that humans may benefit from non-conventional companion animals is through their unique care requirements, which require regular and specific effort to maintain. Stronger adherence to routines necessary for exotic companion animal care may recreate the therapeutic effects of behavioral activation through activity scheduling, for which substantial evidence of its applications in treating depression exist [85,86]. This possibility is yet to be explored in research.

Overall, the body of literature examining the wellbeing benefits of non-traditional companion animals for humans is in its infancy. This review corroborates prior research which identified that descriptive and observational studies dominate human–animal research, and a need for more rigorous and empirical controlled studies [6]. Research on human–animal bonds has previously been noted as “relying on descriptive and correlational evidence” and having a “need for more rigorous empirical studies”, and the sub-field of bonds with non-conventional companion animals appears subject to the same limitations [6]. The qualitative and small-scale descriptive quantitative studies of non-conventional human–animal bonds provide a strong foundation for further research of the topic. The current need is for more empirically rigorous studies which go beyond descriptive statistics and employ controlled testing using larger and more culturally diverse samples. However, empirically testing the health benefits of companion animals is difficult, as previous authors have outlined (see, for example, Rodriguez et al. [4]). For example, participants cannot be blinded to interventions, and randomization is difficult as participants may not have the finances or living conditions to care for companion animals. In qualitative research, which is an important methodological approach to draw upon as well as quantitative and experimental designs, well designed, longitudinal, ethnographic and observational studies would contribute to existing knowledge.

The relatively low number of studies included and species represented in the research identified for this review reinforces the assertion made by Wilkie and Moore [7] that human–animal research overwhelmingly studies specific animals, such as vertebrates and mammals, to the exclusion of others. Furthermore, exactly which species are included in companion animal studies can be difficult to ascertain because animals other than cats and dogs are often reported in a category labelled only as “other” [39]. Likewise, different subspecies of the same animal may be reported without clarification despite important differences in their roles in a household and popular perceptions. For example, “birds” may indicate both parrots, which can speak and are popularly designated as companion animals, or chickens, which are often kept as livestock. This has the consequence of obfuscating important nuance in relationships and possible unique wellbeing benefits people may derive from bonds with non-conventional companion animals. Future research should specifically report species of companion animals involved, and closely test the relationship between attitudes towards species and nuances of the bonds that exist with those species. There is a greatly variable rate of anthropomorphism both within companion animal species, such as between chickens and parrots, and across species, such as between dogs and fish. Future human–animal research comparing bonds with, and perceptions of, different species could also shed light on the connection between anthropomorphism and social support or other health benefits gained from bonds with companion animals.

### Strengths and Limitations

To the authors’ knowledge, this is the first review of health benefits provided by bonds with non-conventional companion animals. Consequently, this study provides a novel perspective for further inquiry into human–animal bonds with more diverse species. However, there are some limitations inherent in the design of narrative reviews. Only articles written in English were considered for inclusion. This is impactful because perspectives of animals and human–animal bonds are highly culturally bound [56]. Different attitudes, species, and health benefits present or popular in non-English speaking cultures were likely missed. Further, the findings of this review may not generalize to human–animal bonds in other cultures.

The lack of randomized controlled trials is understandable, given that randomly assigning companion animals is costly and potentially inconvenient to participants. Additionally, there are logistical and practical difficulties designing effective comparison and control conditions, and ruling out the placebo effect [49]. However, without higher levels of evidence to supplement individual, cross-sectional and anecdotal data, conclusions based on rigorous methodologies are not possible. The field of human–animal bonds research would benefit from more experimental trials measuring impacts of different species of companion animal on human health. This would allow for future meta-analyses to ascertain the health benefits of such species more definitively, and the mechanisms through which these benefits work.

## 5. Conclusions

The available literature on the health benefits of bonds with household companion animals other than dogs and cats is sparse but shows the potential for measurable improvements to people’s health and animals’ capacity to enrich their personal and social lives. Based on this review, sufficient evidence exists that can support future research efforts using empirically rigorous qualitative and quantitative methods to assess benefits of such animals. Future research may consider using standardized measurements of participants’ physical, psychological, and social health to build more evidence about health benefits that can be compared between and across studies. With a dearth of research into numerous species, high-quality qualitative research is necessary, to identify areas for investigation and to contribute to evidence. Any future research should be conducted with animal welfare as a focus, in addition to the benefits for humans. As Shapiro has argued, interdisciplinary, collaborative approaches will likely yield the best research designs and thus, outcomes. Overall, diversifying research into species such as reptiles, insects, fish, birds, and rodents has the potential to better understand how attitudes and species impact human–animal bonds, and the processes of health benefits that may be derived from them.

## Figures and Tables

**Table 1 animals-13-00028-t001:** Peer-reviewed studies included in the review listed in alphabetical order.

Author/s and Date	Study Design	Sample Size	Species	Findings	Themes/Benefit	Country
Azevedo et al. (2022) [38]	Mixed-methods, online survey	220	Reptiles (chelonians, lizards, snakes)	Participants described varying levels of attachment to companion reptiles, ranging to “like” and “love”. Participants also situated companion animals as the focus of pastimes or intellectual curiosity, describing them using terms such as “fascination” and passion”. Most participants (64%) saw their reptile companions as family members, although snakes were less often viewed as family members. Most participants saw reptiles as having the ability to feel stress and fear (80%), and pain and discomfort (74%).	Companionship and Attachment Connectedness to Nature Purpose and Routine	Portugal
Altman (1988) [39]	Narrative review/editorial	N/A	Birds	Synthesizes contemporary research of companion birds to conclude that companion birds should be analyzed with the same attention as cats/dogs	Companionship and Attachment	N/A Articles reviewed written in English
Anderson (2003) [40]	Mixed methods survey circulated over the internet to people with companion parrots	N = 114	Birds (parrots, other)	Companion parrots are capable of complex speech and interaction, prompting deep attachment with humans	Companionship and Attachment	USA
Anderson (2014) [41]	Mixed methods. Ethnographic observation at an avian veterinary clinic, mixed methods surveys	2.5 weeks of fieldwork, 100 questionnaire respondents	Birds	Companion birds as family members; infantilization and anthropomorphism of birds can lead to birds experiencing physical and mental illness, companion birds as objects	Companionship and Attachment Decoration and Aesthetic	USA
Beck and Katcher (1989) [42]	Mixed methods. Standardized questionnaire and some additional questions from Pet Attachment Survey (n = 42) and systematic interviews (n = 18) Qualitative observations of taped systematic interviews	18 interviews, 42 survey respondents	Birds	Birds’ smaller stature requires that the human reduce their level of arousal excitement when interacting with their bird/s, producing a calming effect; dialogue creates companionship; Calming/soothing effect of birds as visual stimulus	Companionship and Attachment	USA
Blecha and Leitner (2014) [43]	Ethnographic approach; participant-observation In-depth interviews and observations with members of eight chicken-keeping households	N = 8 in-depth interviews	Birds (chickens)	Watching and spending time with chickens is stress-relieving Home chicken eggs are considered healthier than store bought eggs Restorative capacities of chickens related to same benefits found from being in nature	Connectedness with Nature Purpose and Routine Decoration and Aesthetic Social Facilitators	USA
Burghardt (2017) [44]	Editorial	N/A	Reptiles and amphibians	Intellectual and academic rewards of keeping reptiles and amphibians. Risks of companion reptiles and amphibians are overstated	Connectedness with Nature	Published in England
Burmeister et al. (2020) [45]	Development and testing of an Owner-Bird Relationship Scale	N = 1444	Birds (parrots and parakeets, finches, ornamental fowl, and “others”)	Anthropomorphism the most important dimension describing the human–bird relationship	Companionship and Attachment Diet, Exercise, Routine Facilitators	Germany
Eddy (1996) [28]	Case study	N = 1	1 snake (common boa constrictor)	Participant’s blood pressure lowered when observing and lowest when interacting with companion snake of 15 years	Companionship and Attachment	Not reported
Kampfer and Love (1998) [46]	Development of a questionnaire, using a survey, named the Tortoise Caregiver Questionnaire	N = 490	Desert tortoises	Tortoises kept for reasons relating to duty, aesthetic, sharing, anthropomorphism, entertainment, relaxation, companionship, feeling needed, providing an escape, and interaction	Companionship and Attachment Connectedness with Nature Decoration and Aesthetic Purpose and Routine	Nevada, USA
Kidd and Kidd (1998) [47]	Interviews and questionnaire	N = 100	Birds	Birds provide friendship, physical and verbal companionship, verbal interactions.	Companionship and Attachment	California, USA
Kidd and Kidd (1999) [48]	Interviews and questionnaire	N = 100	Fish	Calming, relaxation, and stress-reduction effects of watching fish, lessened anxieties, creation of a sense of serenity	Companionship and Attachment Connectedness with Nature Social Facilitators	California, USA
Ko et al. (2016) [49]	Single-Blind Randomized Control Trial using an 8 week pre-post test period	N = 94 46 experimental, 48 control	Insects (crickets)	Caring for insects reduced Geriatric Depression Scale scores and Mini Mental State Examination scores, but not inflammation Participants reported crickets were beneficial for psychological and physical health	Purpose and Routine Connectedness with Nature	Korea
Langfield and James (2009) [50]	Qualitative, phenomenological approach; in-depth semi-structured interviews	N = 9	Fish	Keeping fish provides a meaningful occupation, purpose and enjoyment in life Aquaria act as decorations and improve ambience in homes	Purpose and Routine Decoration and Aesthetic Companionship and Attachment	Newcastle, Australia
Loughlin and Dowrick (1993) [51]	Surveys	N = 80	Birds	Birds fulfill human social needs, followed to a lesser extent by esteem and cognitive needs Useful framework for conceptualizing psychological needs fulfilled by companion animals	Companionship and Attachment	Alaska, USA
Muldoon et al., 2019 [52]	Survey	N = 6700	Fish, reptiles, amphibians, cats, and dogs	Children with companion dogs were significantly more attached to their dogs than children with companion cats, fish, reptiles, or amphibians. Attachment to companion dogs was associated with higher perceived health, happiness, and communication with their father, and slightly increased life satisfaction, but this effect declined with age. The relationship between attachment to other species and psychosocial benefits did not reach statistical significance.	Companionship and Attachment	Scotland
Park, Ko et al. (2019) [53]	Randomized control trial	N = 35 16 control, 19 insect rearing	Insects (crickets)	Insect rearing group showed positive effects on executive functions and performance improvement	Purpose and Routine	Korea
Riddick (1985) [54]	Non-randomized control trial; participants in experimental condition given aquaria containing fish to keep in their homes	N = 22 7 aquarium, 8 visitor, 7 control	Fish	Decrease in blood pressure in aquarium group; significant positive change in leisure satisfaction; significant improvement in relaxation in aquarium group Fish gave “a reason to get up in the morning”	Social Facilitators Purpose and Routine	Maryland, USA
Welle (2011) [55]	Conference proceedings; review of literature and recommendations	N/A	Birds	In-depth exploration of depth of relationships formed between humans and companion birds over the lifespan	Companionship and Attachment	Washington, USA

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
