# Peer review of "Human Health Benefits of Non-Conventional Companion Animals: A Narrative Review"

_animals, 2022, doi:10.3390/ani13010028_

Round 1

Reviewer 1 Report

This is a very well designed, executed and written paper. It was a pleasure to review. I have a few very minor comments:
-I am not sure if the term confer is being used correctly in the context of this paper, do the authors mean offer?
-line 13 - [a] cat...
-line 116 - specie[']s
-line 117-125 -is there not data for Canada?  Are you positive no other country has data.  This may be too bold of a statement. 
-line 146-154 - What about animal assisted activities? Surely bonds are formed given that these visits can span years in some cases. And what about bonds with service animals. Should this be mentioned in a sentence too if AAT is being mentioned?
-should the lack of attention to animal welfare in this paper be noted in the  limitations section?
-line 288 - check wording as go into the quite "reduce...

Author Response

This is a very well designed, executed and written paper. It was a pleasure to review. I have a few very minor comments:

-I am not sure if the term confer is being used correctly in the context of this paper, do the authors mean offer?

We have changed this to “offer”

-line 13 - [a] cat...

We are unsure what this means?

-line 116 - specie[']s

We think that our original is correct, so we have left this.

-line 117-125 -is there not data for Canada?  Are you positive no other country has data.  This may be too bold of a statement. 

We have re-worded the sentence and added Canada.

-line 146-154 - What about animal assisted activities? Surely bonds are formed given that these visits can span years in some cases. And what about bonds with service animals. Should this be mentioned in a sentence too if AAT is being mentioned?

We have not added any of this as our paper specifically excluded these area.

-should the lack of attention to animal welfare in this paper be noted in the  limitations section?

We have added this important consideration in the Conclusion

-line 288 - check wording as go into the quite "reduce...

We have re-worded this

Reviewer 2 Report

Dear authors,

Thank you for the opportunity to read this interesting and novel review of articles.

The introduction is well researched and various studies and theories are thoroughly presented. 

Methods for the use of the narrative synthesis methodology are well described and appear to be thorough with clear parameters 

Results are comprehensive and each theme is described and discussed for various animals. Overall, the Results section and in particular the section on Companionship and Attachment is possibly too detailed and long and may benefit from a briefer summary for ease of reading. 

Discussion is brief and succinct .

Author Response

Thank you for the opportunity to read this interesting and novel review of articles.

The introduction is well researched and various studies and theories are thoroughly presented. 

Methods for the use of the narrative synthesis methodology are well described and appear to be thorough with clear parameters 

Results are comprehensive and each theme is described and discussed for various animals. Overall, the Results section and in particular the section on Companionship and Attachment is possibly too detailed and long and may benefit from a briefer summary for ease of reading. 

Because the other two reviewers did not have an issue with this, and because the reviewer did not suggest specific cuts, we have left these sections as is.

Discussion is brief and succinct.

Reviewer 3 Report

The authors have provided a very thorough narrative review of the relatively few studies about human health benefits of pet ownership among non-conventional companion species. I enjoyed reading the paper and appreciated the mention of methodological limitations throughout. I have provided some feedback below that I hope will help to improve the paper prior to publication.  

Abstract

-          Please provide the date range of studies included (i.e., up until Oct 2022).

-          Include the number of studies.

Introduction

-          I suggest providing examples of some of the physical and psychological health benefits attributed to pet ownership.

-          Please provide specific numbers of ownership rates of non-conventional animals around the world, where possible.  

-          Lines 91-93: Can you provide a little more information about the specific differences in human health benefits between species? How did kids differ in their interactions with dogs vs. non-conventional companion animals? I think it’s worth recognizing that there are some health benefits associated with conventional companion animals that may not be applicable to non-conventional human-animal relationships. E.g., dogs, cats and horses typically have increased physical interactions, can promote physical activity through dog-walking, horse-riding etc., and can promote social interactions with strangers during dog-walking for example. Many of these benefits rely on the behavioral expressions of animals (e.g., the desire/ability to exercise on-leash) and are not solely determined by the human’s perception of the animal.

Results

-          I recommend including a flow diagram to show how many articles were identified from the different sources and any reasons for exclusion.

-          Table 1 is missing a numerical citation for Muldoon et al., 2019.

-          Line 462-465: Is there any evidence to suggest that these interactions are positive? It’s possible that pet caregiving may be a source of stress within the family if the pets were acquired for the children who now view ownership as a chore.

-          Line 466-468: If these three studies show that companion birds can benefit human health by acting as social facilitators between individuals, should they not be included in the review?

-          Line 561: I think you need to soften the wording here. Given the poor methodological rigor of most these studies, I don’t think we can affirm that “People benefit from the appearance…”

-          It’s important to note the significant risk of bias in the conventional media findings. Although individual people may believe that their pets help to provide mental health benefits, these reports are not scientific so I think it is overstated to say these provide “compelling accounts of the mental health benefits”.    

Discussion

-          Line 799: typo in “”nuance sin”

Author Response

The authors have provided a very thorough narrative review of the relatively few studies about human health benefits of pet ownership among non-conventional companion species. I enjoyed reading the paper and appreciated the mention of methodological limitations throughout. I have provided some feedback below that I hope will help to improve the paper prior to publication.  

Abstract

-          Please provide the date range of studies included (i.e., up until Oct 2022).

Done

-          Include the number of studies.

Done

Introduction

-          I suggest providing examples of some of the physical and psychological health benefits attributed to pet ownership.

Because this is contentious in the literature and worthy of a separate paper (of which there are many already) we have not added this.

-          Please provide specific numbers of ownership rates of non-conventional animals around the world, where possible.  

Alas, we have no idea, these data are not freely/readily available.

-          Lines 91-93: Can you provide a little more information about the specific differences in human health benefits between species? How did kids differ in their interactions with dogs vs. non-conventional companion animals? I think it’s worth recognizing that there are some health benefits associated with conventional companion animals that may not be applicable to non-conventional human-animal relationships. E.g., dogs, cats and horses typically have increased physical interactions, can promote physical activity through dog-walking, horse-riding etc., and can promote social interactions with strangers during dog-walking for example. Many of these benefits rely on the behavioral expressions of animals (e.g., the desire/ability to exercise on-leash) and are not solely determined by the human’s perception of the animal.

We have re-worded these lines, to make the meaning clearer, but again, as above, we have not added more to the paper, as we could write yet another paper on this topic.

Results

-          I recommend including a flow diagram to show how many articles were identified from the different sources and any reasons for exclusion.

We had not done this for the study as this is not mandated convention for narrative reviews. However, we have produced a post-hoc PRISMA diagram.  We feel this is not necessarily adding to the paper, but await advice from you and the Editor, if you would like it be included.

-          Table 1 is missing a numerical citation for Muldoon et al., 2019.

Corrected

-          Line 462-465: Is there any evidence to suggest that these interactions are positive? It’s possible that pet caregiving may be a source of stress within the family if the pets were acquired for the children who now view ownership as a chore.

Thank you, we have added a little more from the study, to explain this.

-          Line 466-468: If these three studies show that companion birds can benefit human health by acting as social facilitators between individuals, should they not be included in the review?

We have slightly reworded this to make clearer that we meant to draw comparisons with studies of other animals.  The three studies did not meet inclusion criteria to be included in the review per se.

-          Line 561: I think you need to soften the wording here. Given the poor methodological rigor of most these studies, I don’t think we can affirm that “People benefit from the appearance…”

Done

-          It’s important to note the significant risk of bias in the conventional media findings. Although individual people may believe that their pets help to provide mental health benefits, these reports are not scientific so I think it is overstated to say these provide “compelling accounts of the mental health benefits”.    

Done

Discussion

-          Line 799: typo in “”nuance sin”

Corrected